Determining dense velocity fields for fluid images based on affine motion

Zhang Zili 1
Li Yan 2
Xiang Kaiquan 2
Wang Jinghong 1 wangjinghong@126.com
1 Department of Computer and Cyber Security, Hebei Normal University , Shijiazhuang , China
2 Shijiazhuang Campus, Army Engineering University , Shijiazhuang , China
Szénási Sándor
Electronic publication date: 2024 Feb 16
Publication date: 2024
Volume: 10
Electronic Location ID: e1810
Received 2023 Aug 17; Accepted 2023 Dec 18
Copyright: © 2024 Zhang et al.
Copyright year: 2024
Copyright holder: Zhang et al.
License: This is an open access article distributed under the terms of the Creative Commons Attribution License, which permits unrestricted use, distribution, reproduction and adaptation in any medium and for any purpose provided that it is properly attributed. For attribution, the original author(s), title, publication source (PeerJ Computer Science) and either DOI or URL of the article must be cited.
License URL: https://creativecommons.org/licenses/by/4.0/

Keywords: Velocity field, Fluid image, Patch matching

Funding: Beijing Municipal Natural Science Foundation 4232020 Hebei Normal University L2022B42 Science and Technology Research and Development Program of Shijiazhuang 211130203A This study was funded by the Beijing Municipal Natural Science Foundation (No.4232020), the Doctoral Research Foundation of Hebei Normal University (No. L2022B42), and the Science and Technology Research and Development Program of Shijiazhuang (No. 211130203A). The funders had no role in study design, data collection and analysis, decision to publish, or preparation of the manuscript.

==============================
In this article, we address the problem of estimating fluid flows between two adjacent images containing fluid and non-fluid objects. Typically, traditional optical flow estimation methods lack accuracy, because of the highly deformable nature of fluid, the lack of definitive features, and the motion differences between fluid and non-fluid objects. Our approach captures fluid motions using an affine motion model for each small patch of an image. To obtain robust patch matches, we propose a best-buddies similarity-based method to address the lack of definitive features but many similar features in fluid phenomena. A dense set of affine motion models was then obtained by performing nearest-neighbor interpolation. Finally, dense fluid flow was recovered by applying the affine transformation to each patch and was improved by minimizing a variational energy function. Our method was validated using different types of fluid images. Experimental results show that the proposed method achieves the best performance.

Introduction

Accurate motion estimation is one of the most challenging research problems in computer vision, especially when the focus is fluid-type motion estimation, e.g., for cloud and smoke. In practice, fluid motion estimation is important for many applications, such as forecasting atmosphere evolution (Corpetti, Mémin & Pérez, 2002; Auroux & Fehrenbach, 2011), identifying storms (Lakshmanan, Rabin & Debrunner, 2003), and detecting flame (Mueller et al., 2013; Wang et al., 2016). However, even with the state-of-the-art methods, estimating a dense flow field of fluid from images that contain both fluid phenomena and non-fluid elements, is still yet to be achieved.

The particle image velocimetry (PIV) technology (Prasad, 2000; Scarano, 2012; Cai et al., 2019a) is the most commonly used method, which can provide non-intrusive quantitative measurement of the velocity fields. To construct a PIV experiment, some small particles are seeded into the flow medium. Then successive image sequences with fluorescent particles are captured by a camera. However, it is not proper for some natural scenes, such as a hurricane or a rippling lake.

Optical flow describes a velocity field in the image plane between two consecutive images along the time domain (Verri & Poggio, 1989). Most methods for estimating optical flow integrate the original optical flow formulation from Horn & Schunck (1981), which is based on brightness conservation and spatial coherence assumptions. These generic methods are typically designed for handling quasi-rigid motions with stable salient features. Considering an image sequence of fluid phenomena is mainly dominated by spatial and temporal distortions of luminance patterns, analyzing such an image sequence is challenging and can be hardly handled by classical HS based methods.

To cope with these problems, various approaches based on fluid dynamics constraints are proposed. Corpetti, Mémin & Pérez (2002) proposed a model based on the equation of continuity from fluid mechanics. The divergence-curl technique, describing spreading and rotation, was used as a smoothness constraint (Corpetti, Mémin & Pérez, 2002; Arnaud et al., 2006). To overcome the constant brightness constraint, Li et al. (2010) proposed a Navier-Stokes potential flow framework to model brightness variations in terms of velocity potential. Since these methods mainly rely on physical based fluid motion constraints, they are not suitable for analyzing images that contain a mixture of both fluid phenomena and non-fluid objects, e.g., boats and non-fluid background.

We propose a new method based on affine motion and patch matching through the best-buddies similarity to solve the problem. An affine motion can reasonably model both rigid and local non-rigid motions within a small region (e.g., 3×3 pixels), therefore, our method obtains a dense set of fluid flows by estimating an affine motion model for each image patch. To get a set of robust matching patches, we proposed a novel method based on the best-buddies similarity to get a reliable sparse set of matching patches. Consequently, a sparse set of affine motion models was obtained using these matching patches. Due to large deformations and cluttered backgrounds, some patches could not initially identify corresponding matching patches. A nearest-neighbor interpolation was then applied to estimate affine motion models for these patches. Subsequently, dense fluid flows were obtained by using a dense set of affine motion models. Finally, to remove noise and improve motion estimation, we minimized a variational energy function by initializing the solution with dense fluid flows. Our contributions are as follows: We propose a novel scheme based on the affine motion model and patch matching to estimate a dense motion field for fluid from images that contain fluid phenomena and non-fluid elements.

We propose a robust patch matching method based on the best-buddies similarity of regions. This effectively addresses the problem that fluid usually undergoes complex motions and contains a large number of similar textures.

We propose a new distance metrics combining color, structure, and location to measure the distance between two patches.

Related work

Horn & Schunck (1981) introduced brightness constancy and spatial smoothness assumptions for optical flow estimation. A large number of works (Brox et al., 2004; Sun et al., 2008; Xu, Jia & Matsushita, 2012; Sevilla-Lara et al., 2014; Wulff, Sevillalara & Black, 2017) have been developed to improve the accuracy of the Horn and Schunck formulation. Sun, Roth & Black (2014) provided an overview of recent developments. These methods generally face many challenges, such as brightness changes, large motions, and discontinuities, which are typically associated with fluid phenomena.

Recently, many works have focused on introducing descriptor matching which is robust to large displacements and motion discontinuities (Brox & Malik, 2011) to estimate optical flow (Wills, Agarwal & Belongie, 2006; Liu, Yuen & Torralba, 2011; Revaud et al., 2015; Hu, Li & Song, 2017; Yang & Soatto, 2017). Liu, Yuen & Torralba (2011) used SIFT descriptors to compute dense correspondence fields between two different scenes that comprise very large displacements. Revaud et al. (2015) introduced an edge-preserving interpolation method known as EpicFlow based on geodesic distance. All these methods used rich descriptors to obtain dense flow fields. However, due to the texture similarity of fluids, without additional constraints, it is difficult to identify sufficient and accurate descriptor matching for estimating a dense optical flow through interpolation. Brox, Bregler & Malik (2009) integrated sparse region matches into a variational approach to guide local optimization, forming large displacement solutions. In this method, regions are constructed by hierarchically segmenting an image. Due to texture similarity, it is difficult to segment an image of fluid into appropriate regions for region matching. Yang & Soatto (2017) proposed a method known as S2F to compute the optical flows of fast moving small objects by multi-scale matching, where slower objects are matched first, followed by faster and smaller ones. A key component of the method is to obtain dense optical flows by performing interpolation from initial sparse matches, which were obtained after a series of tests for a dense flow estimated by using existing baseline optical flow algorithms.

In recent years, some approaches have been proposed to recover the velocity field for fluids, such as clouds, smoke, and ocean waves. Popular approaches add fluid dynamics-based constraints to an energy minimization process. The continuity equation of fluid has been integrated into the basic optical flow solution (Corpetti, Mémin & Pérez, 2002; Nakajima et al., 2003). The divergence-curl equation has been used as a smooth constraint to minimize an energy function (Corpetti, Mémin & Pérez, 2002; Arnaud et al., 2006). These methods are all limited by assuming some pre-defined characteristics of optical flow. To solve the problem of brightness changes, a Navier-Stokes based potential flow framework is proposed in Li et al. (2010) to estimate 3D motion. The method is not compatible with turbulence-type of flows. All these methods are limited by the use of physical models, physical based constraints, or the assumption that the entire image only contains fluid flows. They are not applicable to images containing both fluid and non-fluid elements.

In the deep learning era, recent approaches (Ranjan & Black, 2017; Sun et al., 2018; Hui, Tang & Loy, 2018) have designed different neural modules to infer the flow fields. However, as motion itself would introduce clutter and create unpredictable variations, those approaches are still unreliable for ambiguous images, e.g., occlusions. To alleviate the difficulties, some novel strategies (Teed & Deng, 2020; Luo et al., 2022; Xu et al., 2022) are applied to largely reduce the ambiguity of feature matching. However, these methods need a large number of datasets to train models. Meanwhile, these approaches cannot introduce the spatial information and the fluid feature into handling the velocity field estimation. The particle-image velocimetry (PIV) is one of the key techniques to determine the velocity components of flow fields from particle images. Some deep learning-based approaches (Cai et al., 2019b, 2019a; Lagemann et al., 2022) which are trained by a supervised learning strategy, have also been proposed. However, it is difficult to generate the ground truth of fluid flow from nature images without fluorescent particles.

Chen, Li & Hall (2016) proposed a very different approach, using skeletal matching to characterize smoke motion. This method is not a physical-base method, and it has a limit ability to estimate the flow of smoke, because it is hard to extract skeletons from fluid phenomena in general, such as seas and clouds.

Here, we propose a novel fluid flow estimation approach for images that comprise both fluid phenomena and non-fluid elements. The proposed method combines patch matching and affine motion model to estimate fluid flows. An affine motion model could be sufficient to estimate local fluid flows. For instance, Zhou et al. (2001) used an affine motion model to fit a small area of cloud motion. However, the method needs three corresponding candidates given for each point as the initial step and is limited to using multiple image frames as the input. We improve this by requiring only two image frames to obtain the model. In order to obtain more reliable patch matches, we propose a new-patch matching method by determining best-buddies similarity for fluid textures.

Method

Our method estimates fluid flows by assuming that they exhibit non-rigid motions within a small region according to the motion model described in Zhou et al. (2001). Figure 1 gives an overview of our method. First, we produce a sparse set of patch matching between two images using the best-buddies similarity (Dekel et al., 2015). Second, we compute an affine motion model for each patch in the sparse set of patch matches according to the rotation invariant local binary pattern (RLBP) feature (Ojala, Pietikäinen & Mäenpää, 2000). Third, we perform a densification of the affine motion model by interpolating the sparse set using a nearest neighbor interpolation algorithm. Finally, we compute the final fluid flow by minimizing a variational optical flow energy using dense interpolation as initialization.

Figure 1 Method overview.

Given two input images, the initial patch matches were computed (second block) and an affine motion model was estimated for each patch (third block). The sparse set of affine motions was interpolated to obtain a dense set of affine motions (fourth block). The affine motion models of each patch and a variational energy function were combined to obtain the dense fluid flow (final block). Figure source credit: Pexels, https://www.pexels.com/photo/smokeframe-19813961/.

Initial patch matching

The procedure of initial patch matching aims to extract a sparse set of patch matches, facilitating affine motion model estimation. Typically, fluid images possess very few stable salient features but a large number of similar textures. Such images also exhibit high spatial and temporal distortions of luminance patterns. Nevertheless, since the spatial relationships among fluid particles will not change significantly during a small temporal step, we adapt the best-buddies similarity (Dekel et al., 2015) to obtain an optimal match for a given patch. Originally, the best-buddies similarity identifies the fraction of best-buddies pairs, where two points are the nearest neighbors of each other, between two sets of points. We applied the best-buddies similarity measuring the degree of matching between two regions, R={ri∈ℝd}1M and P={pj∈ℝd}1N, where both ri and pj are some small patches of approximately 3×3 pixels. Such a similarity is defined as:

(1) BBS(R,P)=1min{M,N}∑iM∑jNS(ri,pj).

where M and N are the number of patches contained in regions R and P, respectively. Function S shows whether or not the two patches, ri and pj, are best-buddy pairs (Dekel et al., 2015). If pj is the nearest neighbor of ri in R and vice versa, the pair of patches {ri,pj} is regarded as a best-buddies pair. Formally,

(2) S(ri,pj)={1N(ri,P)=pjandN(pj,R)=ri0otherwise

where N(ri,P)=arg⁡minp∈P⁡d(ri,p), and d(ri,p) is the distance between the two patches ri and p, which is measured by some distance metrics. We combine the RGB values, the structure and the location of each patch to measure the distance between two given patches.

Color distance. To measure the color distance between two patches, each patch (of 3×3 pixels) is represented by a 9-element vector of its RGB values, C. In order to overcome the influence of outliers, we use the 2-Norm of color vector to measure color distance as follows:

(3) Lc(ri,pj)=||Cri−Cpj||22.

Structure distance. Color distance is incapable of evaluating the texture distance of two patches. As illustrated in Fig. 2, the color distance between patches a and b is same as that between patches a and c, regardless of their texture differences. So, we integrate the structures of patches into the distance function.

Figure 2 RLBP based patch structure analysis.

(A–C) Show three different texture patches, A, B and C, respectively. The top row shows the gray image and pixel values of each patch. The bottom row shows the corresponding RLBP codes for each patch.

The RLBP operator (Ojala, Pietikäinen & Mäenpää, 2000) defines a texture descriptor T in a local neighborhood of a monochrome texture image as the joint distribution of the signed differences between the gray value of the center pixel, gc and those of the circularly symmetric neighborhood, gn(n=0,1,…,N−1):

(4) T=t(s(gc−g1),s(gc−g2),…,s(gc−gN−1)

The RLBP operator is an excellent measurement of the spatial structure of a local image texture. If the RLBP values of two patches are the same, these patches are regarded as having similar structures. For example, given RLBP(a)=15, RLBP(b)=255 and RLBP(c)=15, the structures of patches a and c are regarded as similar.

So, we define the structure distance between two patches as:

(5) Ls(ri,pj)=|RLBP(ri)−RLBP(pj)|MAXrlab

where MAXrlab is the maximum of RLBP number, such as 255.

Location distance. As a small region usually exhibits fewer structure changes within a short period of time, if two small regions are similar, the distance between two matching patches should then be small. We therefore take such a distance into account.

For distance estimation, the location of a patch is represented by that of its center pixel, which is a local coordinate within an image region, normalized to between 0 and 1. The location distance is then measured by:

(6) Ll(ri,pj)=||Lri−Lpj||22

Finally, the distance function between two patches is defined as:

(7) d(ri,pj)=Lc(ri,pj)+α⋅Ls(ri,pj)+β⋅Ll(ri,pj)

where α and β are scalar coefficients.

To proceed, the original image I1 is first divided into regions of 3Rw×3Rh pixels. Each region is further subdivided into Rw×Rh distinct patches, where each patch contains 3×3 pixels. Second, for each region of image I1, the optimal matching region from image I2 can be determined by maximizing the best-buddies similarity between the two regions (see Eq. (1)). To improve reliability, a region from I2 can be used to estimate the initial patch matching only if the resulting best-buddies similarity is higher than a given threshold, e.g., 0.8, in our experiments. The procedure for generating initial patch matches is shown in Algorithm 1.

Algorithm 1 Generating sparse matches.

Input: I1, I2, a threshold ς	
Output: a set M of sparse patch matches	
1:   Divide the image I1 into a set of small regions {R1,R2,…,RM}	
2:  for each Ri∈I1 do	
3:   for any region Pj∈I2 do	
4:     set D(Ri,Pj)=BBS(Ri,Pj)	
5:     set PM(Ri,Pj)={(ri,pj)|ri∈Ri,pj∈Pj,S(ri,pj)=1}	
6:   end for	
7:  end for	
8:  Determine the optimal matching set Mc={(Ri,Ri′)|Ri′=arg⁡maxPj∈I2⁡D(Ri,Pj)}	
9:  for each (Ri,Ri′)∈Mc do	
10:  if BBS(Ri,Ri′)>ς then	
11:    M=M⋃PM(Ri,Ri′)	
12:  end if	
13:   end for	

Affine motion model

The expected fine and dense fluid field for a fluid scene can be obtained by estimating a motion model for each patch. Palaniappan et al. (1995) and Zhou et al. (2001) confirmed that the affine motion model is suitable to model small areas of local fluid motion. Therefore, we estimated an affine motion model for each small patch. We use a 2D affine motion model (Andrew, 2003) to fit a local patch motion as follows:

(8) qri′k=Miqrik+Ti

where (ri,ri′)∈M, qrik and qri′k are two matching points in the local patches ri and ri′, respectively. Mi is an affine transformation matrix for the patch ri, and Ti is a translation vector.

To get Mi and Ti, we have to obtain a set of point matches which contains at least six elements for each patch match. If a patch undergoes rotation through fluid motion such as vorticity, it is not suitable for a pair of points at the same location of two matching patches to be considered as a matching pair. Hence, for each patch match (ri,ri′)∈M, both ri and ri′ are rotated using the rotation invariant binary patterns (Ojala, Pietikäinen & Mäenpää, 2000). Two points at the same location of the two patches are then considered as a matching pair of points. Finally, we obtain a set of point matches for each patch match, denoted by:

(9) MPrj={(qrjk,qrj′k)},k=1,2,…,|rj|

where |rj| denotes the number of points in the patch rj.

The transformation matrix Mj and the translation vector Tj can be computed by solving the affine motion model (see Eq. (8)) for a patch rj with common numerical methods. We use successive over-relaxation (SOR) to solve the system. Finally, we obtain a sparse set of affine motions, denoted as AM={(Mi,Ti)}.

Interpolation

The few patch matches were transformed into a dense corresponding field by interpolating a sparse set of affine motion models AM={(Mi,Ti)}.

The distance between two patches is an important factor for computing proper interpolation. We used the Euclidean distance D between patches to measure the distance. As the influence of remote patches is either negligible or harmful to the interpolation, we restricted the patches used in the interpolation at a patch r to its ρ neighbors, denoted as Nρ(r), according to distance D.

If BBS(R,R′) is larger, it shows that the results of patch matches inside this region is more reliable. Therefore, we denoted the value BBS(R,R′) as the reliability of a match (ri,ri′)∈M which is used as a scale element for interpolation.

Combing the distance and reliability elements, an interpolation function is defined as:

(10) MD(r)=∑rk∈Nρ(r),(rk,rk′)∈MK(rk,r)(Mk,Tk)∑rk∈Nρ(r),(rk,rk′)∈MK(rk,r)

where rk∈R, rk′∈P, K(rk,r)=exp(−D(rk,r))+μBBS(R,P) and μ is a weight coefficient. This allows us to estimate a dense set of affine motion models, denoted as AM¯ by Eq. (10), where the sparse set of affine motion models AM as the inputs.

Fluid flow optimization

After interpolation, each patch has an affine motion model describing its motion. We then obtained dense fluid flows by carrying out affine translation according to Eq. (8) for each patch. Since only local information is used to estimate the dense fluid flows within each small patch independently, motion may change drastically between two adjacent patches or may have some noise. Meanwhile, due to the absence of a global description of fluid motion, there may be a high degree of overfitting involved in each small area. Hence, some global constraints can be applied to improve the initial fluid motion estimation.

A variational energy function is used to optimize optical flow. For instance, in Brox et al. (2004), brightness constancy assumption, gradient constancy assumption and smoothness constraint were involved. Also, minimizing a variational energy function with a full-scale dense initial flows as the input, can generate better results compared to the conventional coarse-to-fine scheme (Brox et al., 2004; Sun, Roth & Black, 2014). This is applicable to a scene with boundary overlapping or thin objects (Chen, Li & Hall, 2016; Revaud et al., 2015). These phenomena often occur in fluid motion scenes, as the diffusion of fluid may generate some thin fluid areas in the surroundings.

Hence, the dense field generated by our method is used as the initialization of a variational energy minimization method to improve motion estimation. Let x=(x,y)T, w=(u,v)T and ∇=(∂x,∂y)T. We use an energy functional expressing as:

(11) E(w)=∫Ω{Ψ(||I1(x+w)−I2(x)||2)+κΨ(||∇I1(x+w)−∇I2(x)||2)}dx+γ∫ΩΨ(||∇u||2+||∇v||2)

where both κ=30 and γ=80 are tuning parameters, the three terms on the right hand side of the equation are brightness constancy, gradient constancy, and smoothness constraint, respectively, and Ψ(s) is a penalty function (Brox et al., 2004). According to the calculus of variations, we compute Euler-Lagrange equations by preforming three fixed-point iterations and the flow updates three times iteratively.

Experimental results

In this section, we validated our method by performing qualitative and quantitative evaluations on a synthetic image and three different kinds of real-world images, including smoke, hurricane, and clouds. We compare our method with five competing methods, which are the classic or the state-of-the-art motion estimation methods available to us, including HS (Horn & Schunck, 1981), Classic + NL (Sun, Roth & Black, 2014), LDOF (Brox, Bregler & Malik, 2009), EpicFlow (Revaud et al., 2015), and S2F (Yang & Soatto, 2017).

Qualitative analysis

Synthetic image. It is difficult to obtain the ground truth motion field for a real-world fluid phenomenon, therefore, we first synthesized a test image sequence with the given ground truth flow data. To do this, we applied the advection motion and the vortex motion to generate a motion field, which was smoothed by a Gaussian filter. The top, right-hand images in Fig. 3 shows a complicated velocity field that we generated; this image was used as the ground truth for comparison. The figure presents a visual comparison of different methods (second and third rows) against the ground truth velocity field (middle of top row). In the results, the length of each flow vector indicates the magnitude of velocity. To analyze the results, we highlighted part of the flow vectors of the reconstructed smoke motion in a green box, while highlighting part of the reconstructed non-fluid background in a red box. For readability, we also show enlarged views of both boxes. In order to quantitatively evaluate the estimated fluid flow, we choose the average endpoint error (EPE) which computes the Euclidean Distance of the true flow and the estimated one. The proposed method achieves an EPE score of 5.32 surpassing the previous state-of-the-art methods HS, Classic + NL, LODF, EpicFlow, and S2F by 15.98% (6.17 → 5.32), 9.59% (5.83 → 5.32), 3.0% (5.48 → 5.32), 7.52% (5.72 → 5.32), 1.31% (5.39 → 5.32), respectively.

Figure 3 Comparison of fluid motion estimation on synthetic images.

Smoke figure source credit: Pexels, https://www.pexels.com/photo/smokeframe-19813961/.

Generally, all methods can capture both advection and vortex. However, all methods except ours overestimate motion fields for the non-fluid background, generating an improper velocity field and unexpected motions. The Classic+NL method obtains the worst result, generating significantly large velocities for the non-fluid background. In contrast, our method estimates the motion field of the background as having very low (close to zero) velocities. It implies that our method can more accurately estimate fluid flows from images in terms of both fluid phenomena and non-fluid objects.

Smoke. Figure 4 compares fluid motion estimation results. The two adjacent input smoke images are shown in the top row of Fig. 4 as the first two images. Fluid motion estimation results generated from different methods are depicted in rows two to seven of the figure. Specifically, the resultant images of smoke estimation from those methods are presented in the second column, while the velocity field of the highlighted smoke motion region (indicated by the green boxes) is shown in the first column. By visually comparing the results, the reconstructed smoke motion (done by warping interpolation) of our method closely reproduces the expected result (Fig. 4, second image, first row). For results from other methods, for instance, the flow field generated by the HS method is quite noisy. LDOF (Brox, Bregler & Malik, 2009) overestimates flow changes inside the smoke phenomenon, as smoke does not comprise regular structures. More importantly, we produced significantly better flow field estimation for the non-fluid background region, which is important for some applications, e.g., frame detection.

Figure 4 Comparing fluid motion estimation on two different fluid phenomena, namely smoke and hurricane, as shown in the top row.

Six methods are involved, and their corresponding results are shown between the second and seventh rows. Under each phenomenon, the left and right columns respectively show the generated fluid flows and the warping interpolation frames. Smoke figure source credit: Pexels, https://www.pexels.com/photo/smokeframe-19813961/; Hurricane figure source credit: https://www.pexels.com/zh-cn/photo/19742187.

Hurricane. The second fluid phenomenon we tested was a hurricane scene (Fig. 4). The results are depicted in the last two columns. The results are shown as for the smoke motion. Our method captures complicated flow fields extremely well compared to other methods, as highlighted by both the yellow and red boxes. Specifically, besides being capable of capturing complex patterns of flow field, our generated results are less noisy or distorted. Such that our reconstructed image can highly reproduce the ground truth (top right image).

Clouds. Figure 5 depicts another set of experiment results. Here a dense flow field from two input images showing a scene with a cloudy sky is estimated with some rigid moving objects with moving ones, i.e., boats. The first row shows two input images (columns 1, 3) and two of their zoomed partial views comprising the object motions (columns 2, 4 with three small images each). Six methods are involved and their results are shown in 1. Here, the generated flow field (left column) and the reconstructed results (right column), which are obtained by warping the first frame image against the generated flow field, are shown for each method. Both the estimated velocity fields and the interpolation (reconstructed) results show that our method can more accurately capture the subtle cloud motion and the large displacement of boats comparing to other methods. Note that all methods, with the exception of ours and LDOF, overestimate the velocity fields of cloud mainly due to brightness change and a large number of similar textures. However, cloud and boat motions cannot be properly captured by the LDOF method because it lacks the proper segmentations for these features.

Figure 5 Motion flow estimation of a cloud scene including some rigid moving objects.

Cloud figure source credit: Pexels, https://www.pexels.com/zh-cn/photo/19741337/.

Quantitative analysis

The interpolation error (IE) (Baker et al., 2007) is used as the quantitative measure in our experiments because it lacks the ground truth for real world fluid phenomena. For image interpolation, we define IE to be the root-mean-square (RMS) difference between the real second frame ( I2(x,y)) and the interpolated image ( Iw(x,y)) by warping the first frame with the flow field result:

(12) IE=[1N∑(x,y)(Iw(x,y)−I2(x,y))2]12

where N is the number of pixels.

Table 1 compares the errors incurred from our method and other five methods for the four fluid phenomena described in our study. Results show that our method incurred less error than other methods.

Table 1 Quantitative measurement on interpolation error.

The best results are marked in bold for better comparison.

IE	Ours	HS	Classic+NL	LODF	EpicFlow	S2F	
Synthetic images	0.035157	0.064166	0.063931	0.045850	0.063417	0.063562	
Smoke	0.017399	0.058569	0.057548	0.040150	0.058507	0.057942	
Clouds	0.011408	0.024921	0.022619	0.011629	0.021974	0.021930	
Hurricane	0.131075	0.154808	0.151310	0.138125	0.149551	0.150956	

We further quantitatively measure the errors between the ground truth I2 and the warping interpolation frames by computing image differences. Figure 6 shows the results of all the methods we have tested against the three fluid phenomena (smoke, clouds, and hurricane). Our results demonstrate that warping images produced by our method are more consistent with the ground truths than those produced by other methods.

Figure 6 Error analysis on reconstructed fluid images by different methods.

From top to bottom of each column: the ground truth second frame used in computing the fluid flow, the difference images of our method, and five competing methods. Smoke figure source credit: Pexels, https://www.pexels.com/photo/smokeframe-19813961/; Hurricane figure source credit: https://www.pexels.com/zh-cn/photo/19742187; Cloud figure source credit: Pexels, https://www.pexels.com/zh-cn/photo/19741337/.

Conclusion

We have proposed a new approach based on the affine motion model to estimate fluid flow from images, which comprises both fluid phenomena and non-fluid elements. Although fluid typically exhibits nonrigid motions, we demonstrated that we can still use an affine motion model to represent the local motions of fluid. This approach is particularly important because we can represent non-fluid objects properly at the same time within an image. The experimental results have demonstrated the effectiveness of our method.

Our technique has several advantages comparing with existing fluid-based optical flow methods. First, our method obtains a robust patch matching set for fluid phenomena without stable salient features by using the best-buddies similarity based method. Second, we overcame the brightness variation problem. Third, we were able to capture the both subtle fluid motions and large displacements of small rigid objects. Finally, our method can estimate more accurate fluid flows from images including fluid phenomena and non-fluid elements, comparing to existing methods.

For future work, we are interested in extending our method to estimate three dimensional fluid flows which is useful for scene reconstruction and fluid animation. Due to searching for the best-buddy for each small patch from the entire image space, the computational costs of the proposed approach are high. It takes about 4 s to estimate the velocity field for a 560 × 480 image using the Matlab-based implementation. Hence, how to improve the computational efficiency of the algorithm to adapt to online application is another important research topic in the follow-up work. Since the structures of fluid phenomena will change largely over time, the second important future work is to extend our method to obtain accurate fluid flows for a long sequence.

Additional Information and Declarations

Competing Interests

Author Contributions

Data Availability

The authors declare that they have no competing interests.

Zili Zhang conceived and designed the experiments, performed the experiments, analyzed the data, performed the computation work, prepared figures and/or tables, authored or reviewed drafts of the article, and approved the final draft.

Yan Li performed the experiments, analyzed the data, performed the computation work, prepared figures and/or tables, and approved the final draft.

Kaiquan Xiang performed the experiments, authored or reviewed drafts of the article, and approved the final draft.

Jinghong Wang conceived and designed the experiments, analyzed the data, authored or reviewed drafts of the article, and approved the final draft.

The following information was supplied regarding data availability:

The data and code are available at Zenodo: Zhang,. zili. (2023). Determining Dense Velocity Fields for Fluid Images Based on Affine Motion. Zenodo. https://doi.org/10.5281/zenodo.10205205.

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
