# Peer review of "Determining dense velocity fields for fluid images based on affine motion"

_PeerJ Computer Science, doi:10.7717/peerj-cs.1810_

## Round 0.1 · original submission · Major Revisions

Please address the issues raised by the reviewers, especially the problems about the source code. In the case of recommended references, please check the suggestions, but it is optional to follow these.

Reviewer 1 ·

Basic reporting

Please see attached.

Experimental design

Please see attached.

Validity of the findings

Please see attached.

Additional comments

Please see attached.

Annotated reviews are not available for download in order to protect the identity of reviewers who chose to remain anonymous.
Cite this review as

·

Basic reporting

1. Clear and unambiguous, professional English used throughout. (3/5) marks. In random grammar check, I found some issues. I suggest the author re-run the file with some grammar checking software like quill Bot or any of their wish.
2. Literature references, sufficient field background/context provided is sufficient. (4/5). At the end a tabular comparison of existing research would have improved the content.
3. Professional article structure, figures, tables. is good. we can suggest (4/5)
4. Unable to verify the results! I suggest the author resubmit the paper with an updated readme file. Also, I suggest the author that the code should clearly distinguish the Methods, Results proposed in the paper.
5. Unable to verify the condition "Formal results should include clear definitions of all terms and theorems, and detailed proofs." due to the point 4.

Experimental design

1. The article meets the scope.
2. The codebase should have updated read me file. I am unable to run the code and retry the experimentation.
3. Unable to verify the results! I suggest the author resubmit the paper with an updated readme file. Also, I suggest the author that the code should clearly distinguish the Methods, Results proposed in the paper.

Validity of the findings

Unable to verify the results! I suggest the author resubmit the paper with an updated readme file. Also, I suggest the author that the code should clearly distinguish the Methods, Results proposed in the paper.

Novelty can be accepted only after verifying the codebase.

Conclusion should also speak the outcome of the experiment. It should carry the qualitative and quantitative research outcomes.

Additional comments

The readme file in github should include
1. How to run the code.
2. What is the Ideal version of environment, IDE to be used.
3. If you have any video demo of how to do a codesetup in that gitlab. That would be a value added advantage.
4. A pre-recorded result video would be helpful to crosvalidate my result with the result given by the author.

Cite this review as

Reviewer 3 ·

Basic reporting

1. The authors give a detailed demonstration of HS based methods. However, for estimation of fluid-type motion from images, the cross-correlation method is widely used in the community of fluid mechanics. I highly suggest the authors to include the cross-correlation method in their introduction.
Related references:
‘Stereoscopic particle image velocimetry’, A. K. Prasad, 2000.
‘Tomographic PIV: principles and practice’, F. Scarano, 2013.

2. Researches combining optical flow/cross-correlation method and deep learning have been done recently, such as:
‘Particle Image Velocimetry Based on a Deep Learning Motion Estimator’, Cai et al, 2019.
‘LiteFlowNet: A lightweight convolutional neural network for optical flow estimation’, Hui et al, 2018.
I would recommend that the authors mention them in the section of related work, since AI related researches are discussed from line 95 to line 101.

Experimental design

no comment

Validity of the findings

1. The manuscript talks about estimation of flow motion, but there is no direct validation of the velocity estimation throughout.
Therefore I suggest that in figure 3, since the ground truth (i.e., the given motion field) is known, a quantitative comparison between the estimated velocity and the given motion velocity should be made, such as standard deviation of the velocity error.

Additional comments

1. From line 47 to line 53, exact numbers of Sections are missing in the brackets.

2. For the given example on line 131, RLBP(a)= RLBP(c)=15. But it is claimed that a and b are similar structures. Why?

Cite this review as

---

## Round 0.2 · Minor Revisions

Please address the raised minor issue.

Reviewer 1 ·

Basic reporting

no comment

Experimental design

no comment

Validity of the findings

no comment

Additional comments

no comment

Cite this review as

Reviewer 3 ·

Basic reporting

I think the authors have made an effort at addressing the concerns.
The authors claimed that: 'the EPE which computes the european distance of the true flow and the estimated one' (line 210-211). What is the european distance? Do you mean Euclidean Distance?
The paper will be accetable to me if the above concern is resolved.

Experimental design

no comment

Validity of the findings

no comment

Additional comments

no comment

Cite this review as

---

## Round 0.3 · accepted · Accept

The authors have addressed all the issues raised by the reviewers.